# Protective Effect of Inactivated COVID-19 Vaccines against Omicron BA.2 Infection in Guangzhou: A Test-Negative Case-Control Real-World Study

**DOI:** 10.3390/vaccines11030566

**Published:** 2023-03-01

**Authors:** Dingmei Zhang, Jiayi Zhong, Husheng Xiong, Yufen Li, Tong Guo, Bo Peng, Chuanjun Fang, Yan Kang, Jinlin Tan, Yu Ma

**Affiliations:** 1Department of Epidemiology, School of Public Health, Sun Yat-sen University, Guangzhou 510080, China; 2Institute of Public Health, Guangzhou Medical University & Guangzhou Center for Disease Control and Prevention, Guangzhou 510440, China; 3Microbiology Laboratory, Shenzhen Center for Disease Control and Prevention, Nanshan District, Shenzhen 518055, China; 4Department of Public Health and Preventive Medicine, School of Medicine, Jinan University, Guangzhou 510632, China; 5NMPA Key Laboratory for Quality Monitoring and Evaluation of Vaccines and Biological Products, Guangzhou 510080, China

**Keywords:** Omicron infection, vaccination status, protective effect, inactivated COVID-19 vaccines

## Abstract

This study aims to explore the relationship between the doses of inactivated COVID-19 vaccines received and SARS-CoV-2 Omicron infection in the real-world setting, so as to preliminarily evaluate the protective effect induced by COVID-19 vaccination. We conducted a test-negative case-control study and recruited the test-positive cases and test-negative controls in the outbreak caused by Omicron BA.2 in April 2022 in Guangzhou, China. All the participants were 3 years and older. The vaccination status between the case group and the control group was compared in the vaccinated and all participants, respectively, to estimate the immune protection of inactivated COVID-19 vaccines. After adjusting for sex and age, compared with a mere single dose, full vaccination of inactivated COVID-19 vaccines (OR = 0.191, 95% CI: 0.050 to 0.727) and booster vaccination (OR = 0.091, 95% CI: 0.011 to 0.727) had a more superior protective effect. Compared with one dose, the second dose was more effective in males (OR = 0.090), as well as two doses (OR = 0.089) and three doses (OR = 0.090) among individuals aged 18–59. Whereas, when compared with the unvaccinated, one dose (OR = 7.715, 95% CI: 1.904 to 31.254) and three doses (OR = 2.055, 95% CI: 1.162 to 3.635) could contribute to the increased risk of Omicron infection after adjusting for sex and age. Meanwhile, by contrast with unvaccinated individuals, the result of increased risk was also manifested in the first dose in males (OR = 12.400) and one dose (OR = 21.500), two doses (OR = 1.890), and a booster dose (OR = 1.945) in people aged 18–59. In conclusion, the protective effect of full and booster vaccination with inactivated COVID-19 vaccines exceeded the incomplete vaccination, of which three doses were more effective. Nevertheless, vaccination may increase the risk of Omicron infection compared with unvaccinated people. This may result from the transmission traits of BA.2, the particularity and stronger protection awareness of the unvaccinated population, as well as the ADE effect induced by the decrease of antibody titers after a long time of vaccination. It is crucial to explore this issue in depth for the formulation of future COVID-19 vaccination strategies.

## 1. Introduction

In December 2019, a case of pneumonia caused by severe acute respiratory syndrome (SARS-CoV-2) infection was observed in Wuhan, China, and subsequently, the virus spread widely around the world [1]. The disease known as coronavirus disease 2019 (COVID-19), is an emerging infectious disease and has been in existence for almost three years so far [2,3].

During the epidemic, the mutations of SARS-CoV-2 undergo continually. In particular, the global prevalence of Beta, Delta, and Omicron variants poses a threat to the lives and property of people globally [4]. In early November 2021, the Omicron variant of SARS-CoV-2, also called B.1.1.529 [5], was first identified in South Africa [6], and was reported to the World Health Organization (WHO) on 24 November and subsequently listed as a variant of concern (VOC) [7].

The best strategy for achieving herd immunity, which is the most effective method for preventing the COVID-19 epidemic, is the development of safe and effective COVID-19 vaccines. COVID-19 vaccines have been produced in a variety of ways, including DNA-based or mRNA-based vaccines, viral vector-based vaccines, inactivated virus vaccines, attenuated vaccines, and recombinant protein subunit vaccines [8]. Studies have shown that the efficacy of the mRNA-1273 vaccine in phase 3 clinical trials is 94.1% [9], and the efficacy of Pfizer’s vaccine is 52% after the first dose and 95% after the second dose [10]. The results of three randomized, placebo-controlled clinical trials conducted in Brazil, Indonesia, and Turkey all showed an efficacy of more than 50% for the inactivated vaccine. A study from Chile [11] showed an adjusted vaccine effectiveness of 65.9% among people who received two doses. However, the effectiveness of these COVID-19 vaccines was based on the wild strain of SARS-CoV-2.

Both structural modeling and pseudovirus experiments revealed that T478K [12], N501Y and D614G [13,14] mutations in the RBD region of Omicron could increase the tightness and affinity of binding to the ACE2 receptor, which increased the possibility of its transmission. Consequently, the Omicron variant showed 10 times more transmissible than the wild type, and more easily evaded immunization than the Delta variant [15]. The doubling times of cases with Omicron infection were estimated to be 1.2 days, indicating a faster transmission rate [14]. On this basis, several studies suggested that the Omicron variant reduced the effectiveness of COVID-19 vaccines. Compared to the original strain, serum neutralization assays showed a significant 41.4-fold decrease in neutralizing activity against Omicron after two doses of the Pfizer vaccine [16]. Similarly, two doses of the ChAdOx1 vaccine proved to be ineffective against the Omicron variant [17]. Moreover, the vaccine efficacy decreased and antibody levels continued to decline over time [18,19].

Therefore, nowadays, many countries have widely implemented booster vaccination of COVID-19 vaccines to further increase the antibody titers of the Omicron variant in humans and to build an immune barrier to contain the spread of the epidemic. On 8 April 2022, an outbreak of COVID-19 caused by the BA.2 lineage of the Omicron variant occurred in Guangzhou, China. Before the Guangzhou outbreak, most individuals had already received a third dose of COVID-19 vaccines in response to the call. Hence, we explore the relationship between the vaccination status of inactivated COVID-19 vaccines and SARS-CoV-2 Omicron infection through a test-negative case-control design.

## 2. Materials and Methods

### 2.1. Study Design

In this study, a test-negative case-control design was adopted to explore the relationship between different vaccination statuses against COVID-19 and Omicron variant infection in the real-world setting of Guangzhou. For the outbreak on 8 April 2022 in Guangzhou, we included the study participants aged 3 years and older as COVID-19 vaccines have been largely open to all age groups except for infants (under 3 years old) in China since 2021. In addition, booster vaccination has been fully promoted nationwide in China since mid-October 2021.

Study participants were divided into a case group that tested positive for SARS-CoV-2 nucleic acid testing and a control group that tested negative. On the one hand, among the vaccinated participants, three different comparisons of vaccination doses (1 dose versus 2 doses, 1 dose versus 3 doses, 2 doses versus 3 doses) in two groups were analyzed. On the other hand, different doses compared with the unvaccinated individuals were further analyzed, respectively, so as to comprehensively discuss the relationship between vaccination status and Omicron infection. Considering that all data collected in this study were from routine observations, this type of work does not require ethical review, and therefore ethical approval was waived.

### 2.2. Data Sources and Information Collection

All data in the study were obtained from the Guangzhou Center for Disease Control and Prevention (GZCDC). Generally, if the SARS-CoV-2 nucleic acid testing is positive, the case will be centrally isolated immediately and the epidemiological investigations need to be completed in cooperation with GZCDC, which includes verification of activity trajectories, identification and tracing of close contacts and genetic sequencing of the collected respiratory samples to determine the transmission chain where they belong. CDC staff then screened residents in and around the area where the cases lived, as well as exposed persons involved in the epidemic, to further identify potential COVID-19 cases.

Based on these epidemic prevention and control work, researchers collected information on demographics (sex and age), epidemiological history (classification of cases, risk level of infection in close contacts, frequency of exposure, date of symptom onset or last exposure), and COVID-19 vaccination history (type of vaccines, vaccination status and injection date of each dose).

The type of vaccines the participants inoculated in this study consisted of an inactivated vaccine, recombinant subunit vaccine, and adenovirus vaccine (only administered in the control group). The inactivated vaccine is the most widely administered type among study participants, mainly manufactured by Sinovac Research & Development Co., Ltd. (Sinovac R&D) in Beijing, China and China National Biotec Group (CNBG). A few people are vaccinated with recombinant subunit vaccine and adenovirus vaccine. It is hard to estimate the relationship between the two vaccines and Omicron infection. Therefore, this study will focus on inactivated vaccines, of which one dose, two doses, and three doses are defined as incomplete vaccination, full vaccination, and booster vaccination, respectively. According to the vaccination strategy in China, it is suggested that the first dose of inactivated COVID-19 vaccines should be inoculated with the second dose after at least 21 days. Moreover, the third dose of inactivated COVID-19 vaccines, namely a booster dose, from the original or different manufacturers is recommended after the second dose is vaccinated after 6 months or more. In addition to vaccination doses and intervals, booster vaccination also differs from full vaccination in the stronger antibody endurance and wider antibody spectrum.

Furthermore, two weeks (14 days or more) were required to develop adequate protection against SARS-CoV-2 infection. Therefore, we stipulate that the interval between the vaccination of each dose and the symptom onset or last exposure should be at least 14 days. For example, if a participant received the first dose of vaccine within 14 days before infection (for cases) or last exposure (for controls), he or she was considered unvaccinated, and so do the second or third dose.

### 2.3. Selection of Case and Control Groups

In this case-control study, the case group included confirmed cases and asymptomatic cases, and the control group was selected from close contacts of the patients. The cases involved patients who were diagnosed with COVID-19 at medical institutions in Guangzhou from 8 April to 27 April 2022 and met the diagnostic criteria of a positive result for SARS-CoV-2 nucleic acid detection, which was realized by the reverse transcription-polymerase chain reaction (RT-PCR). In this process, viral ribonucleic acid (RNA) needs to be reversely transcribed into complementary deoxyribonucleic acid (cDNA) first, and then the gene sequence is amplified to detect real-time fluorescence or quantitative calculation with special instruments. The GZCDC sequenced the respiratory specimens collected from the cases, and the final results showed that all strains isolated were Omicron BA.2 variants.

It should be emphasized that in accordance with the 8th edition of the Prevention and Control Program implemented in China in April 2022, close contacts are deemed as persons who have frequent exposure to suspected or confirmed cases starting 2 days before the symptom onset or sampling of specimens from asymptomatic cases but have not taken effective protection. In order to exclude infection, once close contacts were identified, 14 days of centralized isolation for medical observation and 7 days of isolation at home for health monitoring (abbreviated as “14 + 7” management measures) should be implemented immediately within 12 h. During this period, repetitive nucleic acid detection was performed on the 1st, 4th, 7th, and 14th days of centralized isolation and the 2nd and 7th days of isolation at home. The control measures can be lifted until the results are persistently negative during the observation period.

During the Omicron outbreak in Guangzhou, there were 303 COVID-19 cases and 45,191 close contacts in total. To ensure comparability between cases and controls and avoid selection bias as much as possible, we subsequently selected 379 close contacts who have a history of frequent exposure to the cases and a high risk of infection, since they may be similar to COVID-19 cases in the experiences of exposure to SARS-CoV-2. Moreover, the exclusion criteria for the case group and the control group include (1) the SARS-CoV-2 nucleic acid test of the case retested negative (N = 25); (2) the existence of missing or erroneous data (N = 6); (3) under the age of 3 years (N = 9); (4) not vaccinated with inactivated vaccines (N = 10). After the above screening, 264 test-positive cases and 368 test-negative controls were included in the following analysis. The screening process of the case group and the control group is shown in Figure 1.

### 2.4. Statistical Analysis

Firstly, categorical and continuous variables were compared, respectively, by Chi-squared test and *t*-test. Meanwhile, the logistic regression model was established to calculate the odds ratio (OR) and its 95% confidence interval (CI) to assess the strength of the association between vaccination doses and Omicron infection. Then, we adjusted OR in logistic regression models with respect to age and sex. Finally, according to sex and age (3–17 years old, 18–59 years old, 60 years old, and above), subgroup analysis is conducted to determine whether these factors have an impact on the correlation strength.

Statistical significance was defined as *p* < 0.05. All data were statistically analyzed by SPSS software (version 26.0).

## 3. Results

### 3.1. Demographic and Epidemiological Characteristics of Included Cases and Controls

Concerning all the participants included in the study, the age of the case group was comparable to that of the control group (median: 33.0 years vs. 34.5 years, *p* = 0.313), and 85.6% of cases (N = 226) and 79.1% of controls (N = 291) had an age distribution between 18 and 59 years. Of the patients, men and women accounted for 46.6% and 53.4%, respectively, while in the control group, the proportion was 48.9% and 51.1%. In terms of vaccination status, overall study participants had, respectively, received one (2.1%), two (38.1%), or three doses (47.2%) of inactivated vaccines against COVID-19 before, with 80 participants never unvaccinated (12.6%).

In conclusion, the result of the analysis revealed statistically significant differences in age group and vaccination status between the case and control groups (*p* < 0.05). The demographic and epidemiological characteristics of the two groups are shown in Table 1.

Of 264 cases included in the study, there were 25 patients who had not been vaccinated, of which 60.0% were males and 40.0% were females. Among the vaccinated cases, 45.2% were males and 54.8% were females. There was no significant difference in sex between the vaccinated and unvaccinated patients. Furthermore, the median age of unvaccinated patients was not significantly different from that of vaccinated cases (median: 32.0 years vs. 33.0 years, *p* = 0.733), and most patients in both groups were between 18 and 59 years old (72.0% vs. 87.0%, *p* = 0.066).

In the unvaccinated control group, the proportion of males and females was 56.4% and 43.6%, respectively, while it was 47.6% and 52.4% in the vaccinated control group (*p* > 0.05). Meanwhile, there was a statistically significant difference between the age groups in the unvaccinated and vaccinated controls, with the majority of them distributed in the range of 18–59 years (*p* = 0.012) (Table 2).

The data shown in Table 3 also demonstrated that among the participants who received two-dose vaccination, the symptoms of cases appeared 273 days after the first dose, while the controls had the last exposure to a case at 246 days after the first dose of vaccine (*p* = 0.001). Similarly, among the participants who had full vaccination, symptoms began to develop 236 days after two doses in the case group, and the last exposure of the control group was 215 days after the second dose (*p* = 0.001). These vaccination intervals in participants vaccinated for two doses were significantly different, which were both longer in the case group than those in the control group.

The interval from the first dose of vaccination to the symptom onset (for cases) or the last exposure (for close contacts) was designated as vaccination interval 1. Relatively, the interval from the last dose of vaccination to the symptom onset (for cases) or the last exposure (for close contacts) was specified as vaccination interval 2.

### 3.2. Protective Effect Induced by Inactivated COVID-19 Vaccines

Among participants who had been vaccinated before, there are 239 cases and 313 close contacts. After adjusting for age and sex, compared with incomplete vaccination, both two-dose vaccination (0.191, 95% CI: 0.050 to 0.727) and three-dose vaccination (0.091, 95% CI: 0.011 to 0.727) of inactivated COVID-19 vaccines showed a more superior protective effect, of which a third dose was more excellent. The participants were further analyzed by subgroup with respect to sex and age. Likewise, the results revealed that two doses were more effective than only a single dose in males (0.090, 95%CI: 0.010–0.770). In addition, among individuals aged 18–59, by comparison with merely one dose, full vaccination (0.089, 95% CI: 0.011 to 0.720), and booster vaccination (0.090, 95% CI: 0.011 to 0.723) had a better protective effect (Table 4).

After involving the unvaccinated population, there were 632 participants in total, which included 264 cases and 368 close contacts. After adjusting for age and sex, however, one dose (OR = 7.715, 95% CI: 1.904 to 31.254) and three doses (OR = 2.055, 95% CI: 1.162 to 3.635) could surprisingly contribute to Omicron infection when compared with the unvaccinated (Table 4). The risk of a single dose was even greater, and a similar result was also manifested in the male population (OR = 12.400, 95% CI: 1.367 to 112.463). After age stratification, it was found that after receiving one, two, and three doses of inactivated COVID-19 vaccines, the risk of infection with Omicron could all increase among people aged 18–59.

## 4. Discussion

It is reported that the Omicron variant has traits of faster transmission and immune escape but reduced lethality compared with previously identified variants of SARS-CoV-2 [20,21], which has led to more insidious community transmission and greater pressure for prevention and control. Currently, booster vaccination against COVID-19 has been fully promoted worldwide in response to the Omicron epidemic. Despite several previous COVID-19 vaccines initially demonstrating high efficacy before [18,22], there have been concerns due to the emergence of various SARS-CoV-2 mutant strains. Compared to the previously prevalent beta and delta strains, the effect of COVID-19 vaccines with different doses (especially booster doses) on the prevention of Omicron infection remains unclear. On 8 April 2022, an epidemic outbreak of the Omicron BA.2 variant in Guangzhou, China, provided an opportunity to evaluate the association between the vaccination status of inactivated COVID-19 vaccines (the most widely administered vaccine in Chinese mainland) and Omicron infection in a real-world setting.

Our research demonstrated that the protective effect of full and booster vaccination with inactivated COVID-19 vaccines exceeded only single-dose vaccination. Furthermore, booster vaccination was more effective. Similarly, after stratification for sex and age, compared with one dose, two doses showed a more excellent protective effect in males, as well as two doses and three doses among individuals aged 18–59. During the Omicron epidemic, a single-dose vaccination was no longer sufficient to fight against the emerging mutant strains of SARS-CoV-2, which has been preliminarily confirmed when the Delta variant was dominant [23,24]. Prior studies have also indicated that COVID-19 vaccines showed high effectiveness after full vaccination and booster vaccination [25,26,27], and the third dose had a stronger protective effect [28]. The findings of the literature were helpful to further support our results, proving that compared to incomplete vaccination, current multi-dose vaccination of inactivated COVID-19 vaccines could provide a better protective effect.

In Table 3, we found that in the participants who received two doses, both the vaccination interval 1 and 2 were significantly longer in the case group than in the control group. This indicated that the longer vaccination intervals contributed to a more inferior protective effect of the inactivated COVID-19 vaccines, which may be related to the decrease of antibody titer in the human body after a long time of vaccination. Likewise, these vaccination intervals were also longer in cases than in controls among the participants vaccinated for single doses and three doses, although the results were not statistically significant. It was probably because the sample size of the one-dose population was too small, and the differences in vaccination intervals of 3-dose participants were not as large as those who received two doses.

However, after further analysis of comparing with unvaccinated participants, it was found that COVID-19 vaccination could unexpectedly increase the risk of Omicron infection, specifically in one dose and three doses of vaccination. A study in the Netherlands also suggested that the protective effect induced by various COVID-19 vaccines was not satisfying and even increased the risk of infection with Omicron BA.1 [29]. Additionally, a study of other variants of concerns (VOCs) came to the same conclusion [30].

We noted that, when compared with the unvaccinated participants, merely one-dose vaccination would make people more susceptible to infection, which is also reflected in the male population. Population aged 18–59 were also at increased risk of Omicron infection after one, two, and three doses of the vaccine. These findings may be related to the limited sample size in this study and the viral transmission characteristics of Omicron BA.2. As noted in a Danish study, BA.2 was associated with increased susceptibility to infection in vaccinated people compared to BA.1 [31]. Moreover, under the background of the vigorous COVID-19 vaccination campaign in China, most of the unvaccinated have relatively special conditions, such as contraindications such as underlying diseases, so they do not have the same sample representation as the general population. Meanwhile, people who are not vaccinated may have stronger protection awareness and more comprehensive preventive measures, which may be one of the reasons why they are not susceptible to being infected. However, this phenomenon might also be explained by the antibody-dependent enhancement (ADE) effect. Low concentrations of neutralizing antibodies combining with two types of Fc–γ receptors (FcgRIIA and FcgRIIIA) on immune cells, and other non-neutralizing antibodies could bind to the pathogen, making the pathogen invade host cells more easily, and result in higher infectivity [32]. In the population vaccinated, longer time intervals will result in a decrease in antibody titer and the lower titer of antibodies will lead to ADE effects. So, those vaccinated were more easily infected than those who were not vaccinated. However, compared with those vaccinated with two and three doses, those vaccinated with one dose generated a weaker immune response and a lower antibody titer, which might more easily induce ADE. Therefore, those vaccinated with two and three doses showed less possibility to be infected than those vaccinated with one dose. Hence, it is essential to determine a proper time for a booster vaccination to avoid ADE caused by low antibody titer. Additionally, inactivated COVID-19 vaccines could produce non-neutralizing antibodies, which might bind to SARS-CoV-2 and induce higher infectivity to Omicron. However, such conclusions need further study to be demonstrated. For vaccination strategy in COVID-19, it is crucial to study this issue in depth.

In this test-negative case-control study, the selection of the control group may affect the accuracy of our results. Consequently, we selected close contacts with a high frequency of exposure and high risk of infection, who were identified by the follow-up of GZCDC. Confounding factors such as age and sex were subsequently adjusted by logistic regression and subgroup analysis. Furthermore, the data from this study derived from patients concentrated in Guangzhou and certainly cannot represent the general situation in China or the world. Nevertheless, our research can still provide a preliminary basis for evaluating the relationship between COVID-19 vaccination status and Omicron infection, especially the unexpected conclusion that the vaccination can increase the risk of Omicron infection when compared to the unvaccinated population, which may be an innovative finding in our study. Meanwhile, we also hope to provide some references for similar research in the future and look forward to a more powerful interpretation of the effective data for the results found in this study.

## 5. Conclusions

This study concluded that compared with incomplete vaccination, the full and booster vaccination of inactivated COVID-19 vaccines showed better protective effects, of which a booster dose was more effective. Whereas, when compared with the unvaccinated, COVID-19 vaccination can even increase the risk of infection, which may be due to the transmission traits of BA.2, the particularity and stronger protection awareness of the unvaccinated population, as well as the ADE effect induced by the decrease of antibody titers after a long time of vaccination. In future research, vaccination regimens and priority populations should be further monitored to explore more vaccination strategies for SARS-CoV-2 mutant strains.

## Figures and Tables

**Figure 1 vaccines-11-00566-f001:**
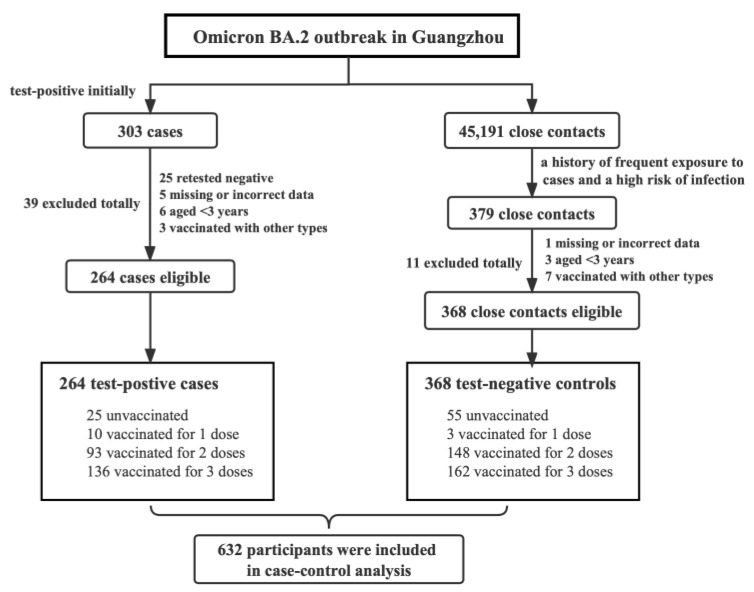
Screening flow chart for study participants. Note: The “unvaccinated” includes those who received their first dose of vaccine less than 14 days prior to infection (N = 2) or last exposure (N = 3), besides those who had never received the COVID-19 vaccine before. The “other types” refer to recombinant subunit vaccine and adenovirus vaccine.

**Table 1 vaccines-11-00566-t001:** Demographic and epidemiological characteristics of case and control groups (*n* = 632).

	Test-Positive Cases (*n* = 264)	Test-Negative Controls (*n* = 368)	*p*-Value
Sex			0.564
Male	123 (46.6%)	180 (48.9%)	
Female	141 (53.4%)	188 (51.1%)	
Age			
Median [IQR]	33.0 [24.0, 43.0]	34.5 [19.0, 46.0]	0.313
Age group (years)			0.026
3–17	25 (9.5%)	62 (16.8%)	
18–59	226 (85.6%)	291 (79.1%)	
≥60	13 (4.9%)	15 (4.1%)	
Vaccination status			0.004
Unvaccinated	25 (9.5%)	55 (14.9%)	
One dose	10 (3.8%)	3 (0.9%)	
Two doses	93 (35.2%)	148 (40.2%)	
Three doses	136 (51.5%)	162 (44.0%)	

**Table 2 vaccines-11-00566-t002:** Demographic characteristics of the case group (*n* = 264) and the control group (*n* = 368).

	Case Group			Control Group		
	Unvaccinated (*n* = 25)	Vaccinated (*n* = 239)	*p*-Value	Unvaccinated (*n* = 55)	Vaccinated (*n* = 313)	*p*-Value
Sex			0.158			0.231
Male	15 (60.0%)	108 (45.2%)		31 (56.4%)	149 (47.6%)	
Female	10 (40.0%)	131 (54.8%)		24 (43.6%)	164 (52.4%)	
Age						
Median [IQR]	32.0 [24.5, 42.5]	33.0 [24.0, 43.0]	0.733	34.5 [24.0, 49.0]	34.5 [18.8, 45.0]	0.129
Age group (years)			0.066 ^1^			0.012
3–17	4 (16.0%)	21 (8.8%)		6 (10.9%)	56 (17.9%)	
18–59	18 (72.0%)	208 (87.0%)		43 (78.2%)	248 (79.2%)	
≥60	3 (12.0%)	10 (4.2%)		6 (10.9%)	9 (2.9%)	

^1^ Fisher’s exact test.

**Table 3 vaccines-11-00566-t003:** Vaccination intervals of case and control groups who had been vaccinated (*n* = 552).

	Vaccinated for 1 Dose			Vaccinated for 2 Doses			Vaccinated for 3 Doses		
	Cases (*n* = 10)	Controls (*n* = 3)	*p*-Value	Cases (*n* = 93)	Controls (*n* = 148)	*p*-Value	Cases (*n* = 136)	Controls (*n* = 162)	*p*-Value
Vaccination interval 1 ^1^									
Median [IQR]	-	-	-	273.0 [240.0, 286.0]	246.0 [152.0, 277.8]	0.001	317.5 [283.0, 333.0]	317.0 [277.0, 337.8]	0.563
Vaccination interval 2 ^2^									
Median [IQR]	234.5 [226.0, 258.5]	155.0 [150.5, 232.0]	0.785 ^3^	236.0 [188.0, 261.0]	215.0 [124.5, 255.0]	0.001	93.0 [48.5, 113.0]	83.5 [35.0, 110.5]	0.562

^1^ Vaccination interval 1: Interval from first dose of vaccination to symptom onset (for cases) or last exposure (for controls). ^2^ Vaccination interval 2: Interval from last dose of vaccination to symptom onset (for cases) or last exposure (for controls). ^3^ In patients who received only one dose, vaccination interval 1 and vaccination interval 2 were the same, and the results were uniformly displayed in the latter.

**Table 4 vaccines-11-00566-t004:** Vaccination status of inactivated COVID-19 vaccines compared in cases and controls against Omicron BA.2 infection in Guangzhou.

	Cases (*n* = 264)	Controls (*n* = 368)	OR_1_ ^3^ (95% CI)	OR_2_ ^3^ (95% CI)	OR_3_ ^3^ (95% CI)
Overall participants ^1^					
Unvaccinated	25 (9.5%)	55 (15.0%)	Reference		
One dose	10 (3.8%)	3 (0.8%)	7.715 (1.904, 31.254) **	Reference	
Two doses	93 (35.2%)	148 (40.2%)	1.474 (0.841, 2.585)	0.191 (0.050, 0.727) *	Reference
Three doses	136 (51.5%)	162 (44.0%)	2.055 (1.162, 3.635) *	0.091 (0.011, 0.727) *	1.069 (0.730, 1.568)
Male ^2^					
Unvaccinated	15 (5.7%)	31 (8.4%)	Reference		
One dose	6 (2.3%)	1 (0.4%)	12.400 (1.367, 112.463) *	Reference	
Two doses	42 (15.9%)	78 (21.1%)	1.113 (0.541, 2.290)	0.090 (0.010, 0.770) *	Reference
Three doses	60 (22.7%)	70 (19.0%)	1.771 (0.874, 3.590)	0.143 (0.017, 1.220)	1.592 (0.956, 2.650)
Female ^2^					
Unvaccinated	10 (3.8%)	24 (6.5%)	Reference		
One dose	4 (1.5%)	2 (0.6%)	4.800 (0.754, 30.550)	Reference	
Two doses	51 (19.3%)	70 (19.0%)	1.749 (0.769, 3.975)	0.364 (0.064, 2.066)	Reference
Three doses	76 (28.8%)	92 (25.0%)	1.983 (0.893, 4.403)	0.413 (0.074, 2.317)	1.134 (0.707, 1.817)
3–17 years ^2^					
Unvaccinated	4 (1.5%)	6 (1.6%)	Reference		
One dose	1 (0.4%)	2 (0.6%)	0.750 (0.050, 11.311)	Reference	
Two doses	20 (7.6%)	54 (14.6%)	0.556 (0.142, 2.176)	0.741 (0.064, 8.624)	Reference
Three doses	0 (0.0%)	0 (0.0%)	- ^4^	- ^4^	- ^4^
18–59 years ^2^					
Unvaccinated	18 (6.8%)	43 (11.7%)	Reference		
One dose	9 (3.4%)	1 (0.4%)	21.500 (2.535, 182.373) **	Reference	
Two doses	72 (27.3%)	91 (24.7%)	1.890 (1.006, 3.553) *	0.089 (0.011, 0.720) *	Reference
Three doses	127 (48.1%)	156 (42.4%)	1.945 (1.069, 3.537) *	0.090 (0.011, 0.723) *	1.029 (0.698, 1.516)
≥60 years ^2^					
Unvaccinated	3 (1.1%)	6 (1.6%)	Reference		
One dose	0 (0.0%)	0 (0.0%)	- ^4^	Reference	
Two doses	1 (0.4%)	3 (0.8%)	0.667 (0.047, 9.472)	- ^4^	Reference
Three doses	9 (3.4%)	6 (1.6%)	3.000 (0.533, 16.897)	- ^4^	4.500 (0.374, 54.155)

^1^ ORs were adjusted for sex and age. ^2^ ORs were unadjusted. ^3^ OR_1_: vaccination status compared with the unvaccinated; OR_2_: vaccination status compared with one dose; OR_3_: vaccination status compared with two doses. ^4^ The “-” indicates no data or cannot be analyzed due to the insufficient sample size in this subgroup. * *p* < 0.05. ** *p* < 0.01.

## Data Availability

Not applicable.

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
