# Peer review of "Protective Effect of Inactivated COVID-19 Vaccines against Omicron BA.2 Infection in Guangzhou: A Test-Negative Case-Control Real-World Study"

_vaccines, 2023, doi:10.3390/vaccines11030566_

Round 1
Reviewer 1 Report
The MS described a case study on efficacy of vaccination with inactive SAR-COV-2. It would be a big interest to readers. However, a number of points need to be improved or changed in current version.
1. The results cannot support the conclusion in current version. The results cannot fully conclude that vaccination was helpful to prevention of infection. For example, the results in table 1 showed that the unvaccinated percent present higher levels in case control group than case group. The results in table 4 cannot explain the conclusion yet.
2. The results of Table 3 presented statistically significant for case with 2 doses vaccination but not for 1 or 3 doses vaccination. It is hard to understand about that for readers. I think that it cannot due to limitation of sample numbers.
3. The case numbers were listed in Table4, the case percent should be added in.
4. The writing needs to be modified. Many sentences are hard to be understood or have grammar mistakes. For example, ...developing effective and safe COVID-19 vaccines is the best choice among the means to obtain herd immunity on line 59-61, or Our research showed that compared with single-dose vaccination, both full vaccination and booster vaccination of inactivated COVID-19 vaccines could reduce the risk of contracting Omicron variant on line 266-268.
Author Response
Response to Reviewer 1 Comments
Point 1: The results cannot support the conclusion in current version. The results cannot fully conclude that vaccination was helpful to prevention of infection. For example, the results in table 1 showed that the unvaccinated percent present higher levels in control group than case group. The results in table 4 cannot explain the conclusion yet.
Response 1: Thank you for pointing out this problem. Our conclusion that vaccination helps prevent infection is based on comparisons with participants who received one-dose vaccination, rather than the unvaccinated individuals. When compared with unvaccinated participants, COVID-19 vaccination may cause infection, as shown in Table 1. The results in Table 4 show that both two and three doses of vaccination are protective (OR<1 and statistically significant) when compared with one dose. In contrast, when compared with unvaccinated participants, we concluded that vaccination may lead to increased susceptibility to Omicron variant. We modified relevant sentences.
Point 2: The results of Table 3 presented statistically significant for case with 2 doses vaccination but not for 1 or 3 doses vaccination. It is hard to understand about that for readers. I think that it cannot due to limitation of sample numbers.
Response 2: Thank you very much for your suggestion. We have added several sentences and corresponding literature in the discussion section to enrich the content and better explain the results of Table 3: "But this phenomenon might also be explained by antibody-dependent enhancement (ADE) effect.......". The significance of the results is described according to the characteristics of the relationship between the different time intervals of one, two and three doses and the changes in antibody levels (page 9, lines 335-348).
Point 3: The case numbers were listed in Table4, the case percent should be added in.
Response 3: Thank you for your kind suggestion. We have added in Table 4 the percentages of cases and controls in terms of vaccination status. In the stratified analysis for age and sex, the percentages of cases and controls in each stratification add up to 100%, respectively.
Point 4: The writing needs to be modified. Many sentences are hard to be understood or have grammar mistakes. For example, ...developing effective and safe COVID-19 vaccines is the best choice among the means to obtain herd immunity on line 59-61, or Our research showed that compared with single-dose vaccination, both full vaccination and booster vaccination of inactivated COVID-19 vaccines could reduce the risk of contracting Omicron variant on line 266-268.
Response 4: We deeply appreciate the reviewer's suggestion. Based on your suggestions, we have improved the expressions of the sentences (page 2, lines 63-65 and page 8, lines 291-295). In addition, we have further modified the writing of the manuscript.
Reviewer 2 Report
The authors evaluated the relationship between the doses of inactivated COVID-19 vaccines received and SARS-CoV-2 Omicron infection in Guangzhou. They found compared with one dose, the second dose made males less susceptible to infection as well as two doses and three doses among individuals aged 18-59.
Comments for the authors:
Major points:
1. Please explain the difference between full vaccination and booster vaccination.
2. Please indicate the method for PCR testing, if possible.
Author Response
Response to Reviewer 2 Comments
Point 1: Please explain the difference between full vaccination and booster vaccination.
Response 1: Thanks a lot for the reviewer's suggestion. We have added to the Materials and Methods section an explanation on the differences between full vaccination and booster vaccination. To enrich the content, we explain the differences of required vaccination doses, time intervals and duration of antibody level in detail (page 3, lines 140-147).
Point 2: Please indicate the method for PCR testing, if possible.
Response 2: Thank you for your valuable suggestion. We have added a specific introduction of RT-PCR detection technology and its method (pages 3-4, lines 159-163).
Round 2
Reviewer 1 Report
The current version cannot get the inclusion of vaccine efficacy based on the results. The statistical analysis in table 4 presented somewhat differences between the groups. However, compared to unvaccinated group or dose 1 group, the percentage of two or three dose group is the vast majority in both cases and control. I do not understand how the vaccine decreased the infections.
Author Response
Response to Reviewer 1 Comments
Point 1: The current version cannot get the inclusion of vaccine efficacy based on the results. The statistical analysis in table 4 presented somewhat differences between the groups. However, compared to unvaccinated group or dose 1 group, the percentage of two or three dose group is the vast majority in both cases and control. I do not understand how the vaccine decreased the infections.
Response 1: We deeply appreciate your valuable suggestion. Our original intention was to show that the protective effect of two doses and three doses were better than one dose. We would like to emphasize that the vaccine efficacy in this study was determined by the OR value (the odds ratio of the vaccination status in those who are infected and those who are not). If the ratio of vaccination to non-vaccination was higher in the case group than in the control group, the vaccination increased the risk of infection. However, in this study, we concluded that the vaccine increased the infection, and it is obviously not appropriate to conclude that two and three doses’ vaccine can prevent infection when compared to a single dose. Therefore, in order to avoid misunderstanding, we have revised all relevant sentences in the manuscript, specifically in lines 29-33 on page 1, lines 38-40 on page 1, lines 263-271 on page 7, lines 321-333 on page 8, and lines 433-435 on page 9, and we discussed the potential explains for our results (page 9, lines 403-417). Meanwhile, several references were added to enrich the content and better support our results. Thank you very much for pointing out this problem.
Round 3
Reviewer 1 Report
The text of the revised MS have great improvement although the conclusion of vaccine efficacy seems ambiguous. However, the ambiguous conclusion may be clarified in discussion part.
Author Response
Response to Reviewer 1 Comments
Point 1: The text of the revised MS have great improvement although the conclusion of vaccine efficacy seems ambiguous. However, the ambiguous conclusion may be clarified in discussion part.
Response 1: Thank you very much for your valuable suggestion. In response to your comments, we have further clarified our conclusions regarding vaccine efficacy (vaccine effectiveness when compared to the unvaccinated population) in discussion part. We have revised the statement in the last paragraph of the discussion part to highlight the key conclusion of our study: "......especially the unexpected conclusion that the vaccination can increase the risk of Omicron infection when compared to the unvaccinated population, which may be an innovative finding in our study." Thank you for pointing out this problem.